# Repurposing FIB-4 index as a predictor of mortality in patients with hematological malignancies and COVID-19

**Noorwati Sutandyo**[1]*, **Sri Agustini Kurniawati**[1], **Achmad Mulawarman Jayusman**[2], **Anisa Hana Syafiyah**[3], **Raymond Pranata**[4], **Arif Riswahyudi Hanafi**[2]

1 Hematology Cancer Teamwork, Department of Medical Hematology-Oncology, Dharmais National Cancer Center, Jakarta, Indonesia, 2 COVID-19 Mitigation Research Team, Department of Pulmonology, Dharmais National Cancer Center, Jakarta, Indonesia, 3 Hematology Cancer Research Team, Dharmais National Cancer Center, Jakarta, Indonesia, 4 Statistical Analysis Consultant, Indonesia

* noorwatis3@yahoo.com

**Data Availability Statement:** Data cannot be shared publicly because the data contain potentially identifying or sensitive patient information. The Ethical Research Committee of Dharmais National

## Abstract

### Background

In this study, we aimed to investigate whether FIB-4 index is useful in predicting mortality in patients with concurrent hematological malignancies and COVID-19. We also aimed to determine the optimal cut-off point for the prediction.

### Methods

This is a single-center retrospective cohort study conducted in Dharmais National Cancer Hospital, Indonesia. Consecutive sampling of adults with hematological malignancies and COVID-19 was performed between May 2020 and January 2021. COVID-19 screening test using the reverse transcriptase polymerase chain reaction (RT-PCR) of nasopharyngeal samples were performed prior to hospitalization for chemotherapy. FIB-4 index is derived from [age (years) × AST (IU/L)]/[platelet count ($10^9$/L) × $\sqrt{}$ALT (U/L)]. The primary outcome of this study is mortality, defined as clinically validated death/non-survivor during a 3-months (90 days) follow-up.

### Results

There were a total of 70 patients with hematological malignancies and COVID-19 in this study. Median FIB-4 Index was higher in non-survivors (13.1 vs 1.02, p<0.001). FIB-4 index above 3.85 has a sensitivity of 79%, specificity of 84%, PLR of 5.27, and NLR of 0.32. The AUC was 0.849 95% CI 0.735–0.962, p<0.001. This cut-off point was associated with OR of 16.70 95% CI 4.07–66.67, p<0.001. In this study, a FIB-4 >3.85 confers to 80% posterior probability of mortality and FIB-4 <3.85 to 19% probability. FIB-4 >3.85 was associated with shorter time-to-mortality (HR 9.10 95% CI 2.99–27.65, p<0.001). Multivariate analysis indicated that FIB-4 >3.85 (HR 4.09 95% CI 1.32–12.70, p = 0.015) and CRP> 71.57 mg/L (HR 3.36 95% CI 1.08–10.50, p = 0.037) were independently associated with shorter time-to-mortality.

Cancer Hospital, Indonesia imposes strict restriction regarding patients' confidentiality. However, all reasonable data access requests may be sent to the Ethical Research Committee of Dharmais National Cancer Hospital, Indonesia (email: kepkdharmais@gmail.com) or to the address at Jl. Letjen S. Parman No.84-86, Kota Bambu Selatan, Kec. Palmerah, Kota Jakarta Barat, Daerah Khusus Ibukota Jakarta 11420.

**Funding:** The author(s) received no specific funding for this work.

**Competing interests:** The authors have declared that no competing interests exist.

## Conclusion

This study indicates that a FIB-4 index >3.85 was independent predictor of mortality in patients with hematological malignances and COVID-19 infection.

## Introduction

Coronavirus Disease-2019 (COVID-19) manifest with different clinical presentations which might be asymptomatic, mild-moderate, and severe, which may cause multi-organ failure and death [1]. The severity of COVID-19 is affected by the presence of comorbidities [2–4]. During the pandemic, healthcare system is often overloaded with COVID-19 patients and risk stratification is of paramount importance for optimum resource allocation. Liver enzymes and platelets are often altered in patients with hematological malignancies; and might be useful for predicting outcome. Thrombocytopenia in also often encountered in patients with COVID-19 and signifies poor prognosis [5].

Fibrosis-4 (FIB-4) index is derived from the calculation of age, liver enzymes, and platelets, which are often evaluated in patients with hematological malignancies, thus is cost-effective and efficient. Moreover, FIB-4 index has also been shown to be associated with mortality in patients with COVID-19, even in the absence of previously known liver disease [6–8]. To the best of the authors knowledge, this is the first study to describe the use of FIB-4 index in patients with hematological malignancies, with or without COVID-19. In this study, we aimed to investigate whether FIB-4 index is useful in predicting mortality in patients with concurrent hematological malignancies and COVID-19. We also aimed to determine the optimal cut-off point for the prediction.

## Materials and methods

### Study design

This is a single-center retrospective cohort study conducted in Dharmais National Cancer Hospital, Indonesia. Consecutive sampling of adults who were tested positive for COVID19 while undergoing chemotherapy for hematological malignancy was performed between May 2020 and January 2021. The baseline characteristics and admission laboratory values were obtained from medical records. Hematological malignances were defined as patients diagnosed with acute myelogenous leukemia (AML), acute lymphocytic leukemia (ALL), chronic myelogenous leukemia (CML), chronic lymphocytic leukemia (CLL), multiple myeloma (MM), non-Hodgkins lymphoma (NHL), or Hodgkins lymphoma (HL). COVID-19 screening test using the reverse transcriptase polymerase chain reaction (RT-PCR) of nasopharyngeal samples were performed prior to hospitalization for chemotherapy. COVID-19 diagnosis was based on the positive RT-PCR examination. The study was performed in accordance with the ethical standards of the 1964 Helsinki Declaration and its later amendments. Ethical clearance was provided by Ethical Research Committee of Dharmais National Cancer Hospital, Indonesia (Number: 0149/KEPK/X/2020). Ethics committee waived the requirement for informed consent due to retrospective observational nature of the study.

### COVID-19 medications

The patients were given favipiravir (2 x 1600 mg) followed by maintenance (2 x 600 mg), azithromycin 1 x 500 mg, oseltamivir 2x75 mg, vitamin D3 1 x 1000 IU, vitamin C 2 x 500 mg,

zinc 1 x 20 mg, betadine gargle 3 x 15 ml, and iodine nasal spray 3 x 1 for COVID-19 treatment. The risk of potential mechanisms or drug-to-drug interaction that may aggravate cancer progression with the use of these medications is low.

## FIB-4 index

FIB-4 index is derived from [age (years) × AST (IU/L)]/[platelet count ($10^9$/L) × $\sqrt{}$ALT (U/L)]. This variable was calculated in the form of continuous data and categorized based on the optimal ROC curve determined cut-off point and a cut-off value of 1.45, which is commonly used to rule out certain outcomes.

## Outcome

The primary outcome of this study is mortality, defined as clinically validated death/non-survivor during a 3-months (90 days) follow-up. The outcome was ascertained from the medical record confirmed by death certificate. The statistical analysis was performed by a researcher that is not involved in data collection or patient care.

## Statistical analysis

Statistical analysis was performed using SPSS 25.0 and STATA 14.0. The continuous data was tested for normal distribution; t-test was used for normally distributed data and Mann-Whitney test was used for abnormally distributed data. Normally distributed data was reported as mean and standard deviation (SD); while abnormally distributed data was reported as median and interquartile range (IRQ). The continuous variables that significantly differ in non-survivors vs. survivors were then categorized. ROC curve analysis was performed to determine the optimal cut-off points for FIB-4 index and C-reactive protein (CRP). For FIB-4 index, a cut-off value of 1.45 was also analyzed because it is commonly used to rule out poor prognosis. The sensitivity, specificity, positive likelihood ratio (PLR), negative likelihood ratio (NLR), and area under the curve (AUC) were calculated for the respective cut-off points. Fagan's nomogram was plotted to determine the posterior probability of mortality in the event of FIB-4 index elevation above the cut-off point. Chi-square was used to calculate the odds ratio (OR) and its 95% confidence interval (95% CI) if the requirements are met. Cox-regression was performed to obtain the hazard ratio (HR) and survival curve based on the optimal FIB-4 index. Multivariable Cox-regression analysis was initially performed by including FiB-4 index, hemoglobin <10 g/dL, and CRP >71.57 mg/L to find the independent predictors. The number of variables were limited to three for the multivariable analysis to avoid model-overfitting. Multivariable analysis was only performed for HR to minimize the familywise error rate. Subgroup analysis for acute and chronic types of malignancies were performed, we obtained the ORs but not HRs, because chronic subgroup has spuriously high hazard ratio.

## Results and discussion

There were a total of 70 patients with hematological malignancies and COVID-19 in this study. The baseline characteristics of the study can be seen in Table 1. Severe COVID-19 was higher in non-survivors. The mean hemoglobin, platelets, and FIB-4 Index were lower in non-survivors; while CRP was higher in non-survivors. The mortality rate was 43% and the median follow-up length was 33 days (IQR 64).

**Table 1. Baseline characteristics.**

| | Non-Survivor (n = 28) | Survivor (n = 42) | p-value |
|---|---|---|---|
| **Age (years)** | 44.9 ± 12.8 | 42.7 ± 14.6 | 0.339 |
| **Gender (Male)** | 15 (53.6%) | 23 (54.8%) | 0.922 |
| **BMI (kg/m$^2$)** | 22.4 (6.68) | 21.2 (3.65) | 0.172 |
| **Severe COVID-19** | 23 (82.1%) | 4 (9.5%) | <0.001 |
| **Cancer Type** | | | 0.180 |
| **AML** | 10 (35.7%) | 10 (23.8%) | 0.280 |
| **ALL** | 4 (14.3%) | 3 (7.1%) | 0.426 |
| **CML** | 1 (3.6%) | 3 (7.1%) | 0.645 |
| **CLL** | 2 (7.1%) | 2 (4.8%) | 1.000 |
| **MM** | 4 (14.3%) | 2 (4.8%) | 0.209 |
| **NHL** | 7 (25.0%) | 16 (38.1%) | 0.253 |
| **HL** | 0 (0%) | 6 (14.3%) | 0.074 |
| **Rituximab Use in CLL and NHL** | 5 (100%) | 9 (75%) | 0.515 |
| **Remission and Relapse Status** | | | |
| **Remission** | 2 (7.1%) | 3 (7.1%) | 1.000 |
| **Relapse** | 2 (7.1%) | 6 (14.3%) | 1.000 |
| **Neither** | 24 (85.8) | 33 (78.6%) | 0.329 |
| **Proliferation** | | | 0.301 |
| **Fast** | 13 (46.4%) | 12 (28.6%) | 0.127 |
| **Moderate** | 12 (42.9%) | 25 (59.5%) | 0.171 |
| **Slow** | 3 (10.7%) | 5 (11.9%) | 1.000 |
| **Laboratory Values** | | | |
| **Hemoglobin (g/dL)** | 8.5 ± 1.7 | 10.5 ± 2.1 | <0.001 |
| **Leukocyte (x 10$^9$/L)** | 12.6 (15.8) | 14.5 (100.9) | 0.607 |
| **Platelets (x 10$^9$/L)** | 60.0 (573) | 69.0 (334) | <0.001 |
| **Ureum (mg/dL)** | 38 (31) | 32 (28) | 0.011 |
| **Creatinine (mg/dL)** | 0.60 (0.42) | 0.63 (0.52) | 0.317 |
| **AST (U/L)** | 30 (60) | 21 (25) | 0.752 |
| **ALT (U/L)** | 51.5 (77) | 16.0 (38) | 0.239 |
| **FIB-4 Index** | 13.1 (12.1) | 1.02 (2.53) | <0.001 |
| **FIB-4 Index <3.85** | 6 (27.3%) | 16 (72.7%) | <0.001 |
| **FIB-4 Index >3.85** | 25 (86.2%) | 4 (13.8%) | <0.001 |
| **PT** | 17.4 ± 4.6 | 19.8 ± 18.0 | 0.585 |
| **aPTT** | 35.0 ± 13.9 | 52.0 ± 77.2 | 0.297 |
| **D-Dimer (mg/dL)** | 2662 ± 1564 | 2163 ± 1617 | 0.317 |
| **CRP (mg/L)** | 87.1 (123.2) | 28.6 (58.2) | <0.001 |

The survivors and non-survivors are based on 3 months (90 days) mortality.

Continuous variables were presented in mean ± standard deviation or median (interquartile range)

AST: aspartate aminotransferase, ALT: alanine aminotransferase, AML: acute myelogenous leukemia, ALL: acute lymphocytic leukemia, CML: chronic myelogenous leukemia, CLL: chronic lymphocytic leukemia, MM: multiple myeloma, NHL: non-Hodgkins lymphoma, or HL: Hodgkins lymphoma, CRP: c-reactive protein, FIB-4 index: fibrosis-4 index, PT: Prothrombin Time, aPTT: Activated Partial Thromboplastin Time

## FIB-4 index >3.85 and mortality

Median FIB-4 Index was higher in non-survivors (13.1 vs 1.02, p<0.001). Through the ROC curve analysis, the optimal cut-off point for FIB-4 index was determined to be 3.85, which has a sensitivity of 79%, specificity of 84%, PLR of 5.27, and NLR of 0.32. The AUC was 0.849 95%

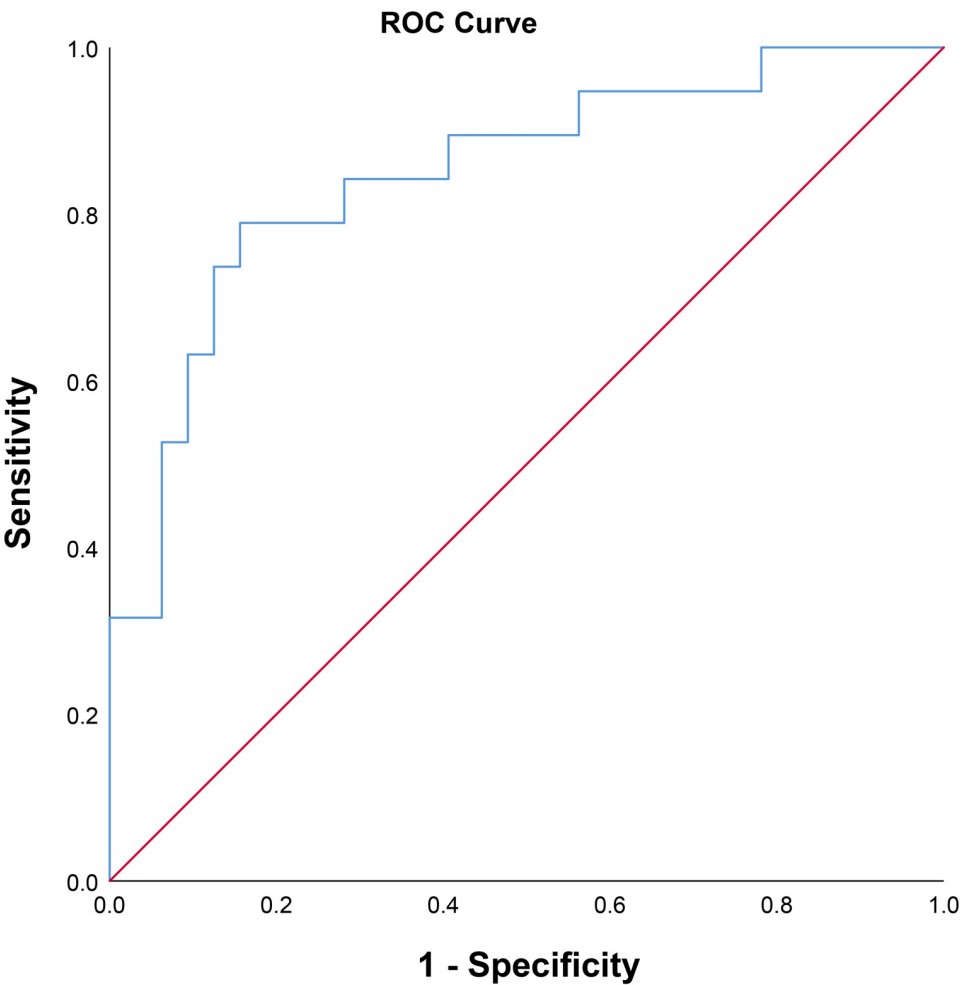

**Fig 1. Receiver operating characteristic curve for FIB-4 and mortality.**

CI 0.735–0.962, p<0.001 [Fig 1]. This cut-off point was associated with OR of 16.70 95% CI 4.07–66.67, p<0.001. In this study, a FIB-4 >3.85 confers to 80% posterior probability of mortality and FIB-4 <3.85 to 19% probability [Fig 2]. FIB-4 >3.85 was associated with shorter time-to-mortality (HR 9.10 95% CI 2.99–27.65, p<0.001) [Fig 3].

## FIB-4 >1.45 and mortality

FIB-4 >1.45 was associated with higher mortality (OR 12.05 95% CI 2.86–50, p<0.001) with a sensitivity of 84%, specificity of 59%, PLR of 2.50, and NLR of 0.21. In this study, a FIB-4 >1.45 confers to 65% posterior probability of mortality and FIB-4 <1.45 to 14% probability [Fig 4]. FIB-4 >1.45 was associated with shorter time-to-mortality (HR 5.43 95% CI 1.58–18.70, p = 0.007) [Fig 5].

## Other markers

Hemoglobin <10 g/dL (HR 8.13 95% CI 2.44–27.03, p = 0.001) and CRP >71.57 mg/L (HR 4.98 95% CI 1.86–13.3, p = 0.001) were associated with shorter time-to-mortality. CRP >71.57

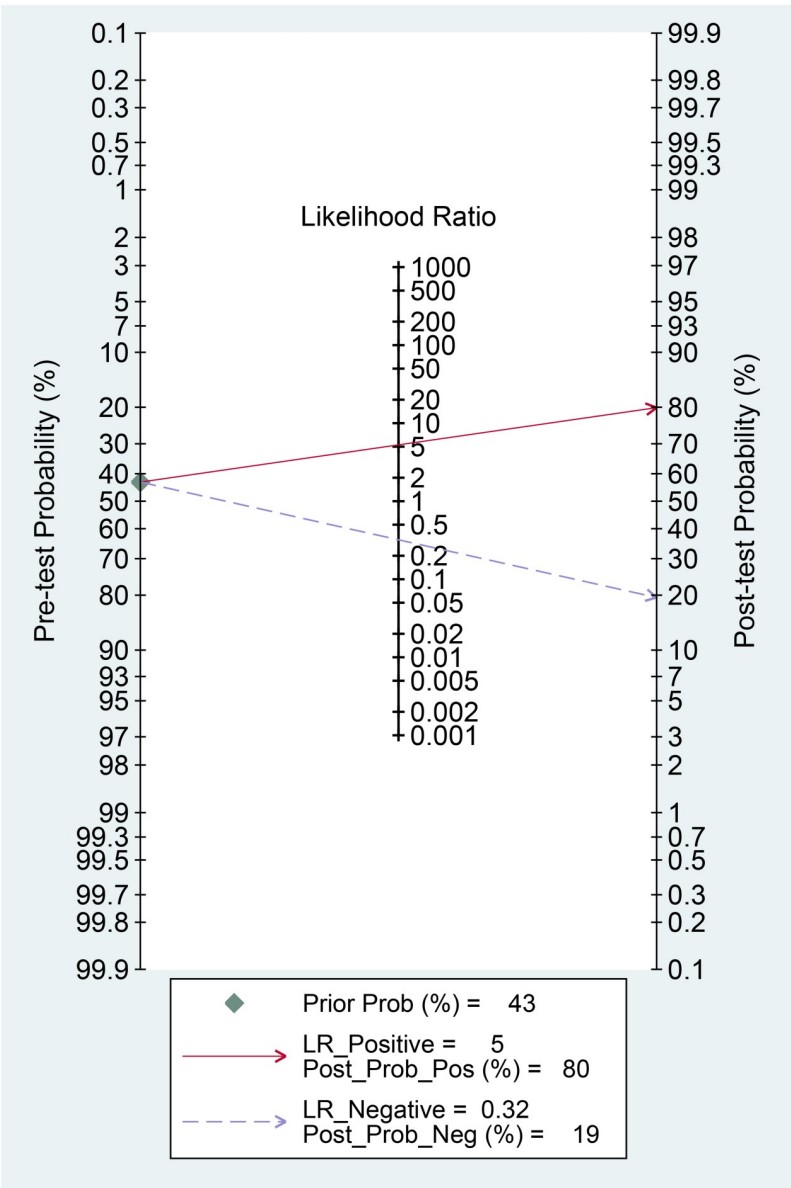

**Fig 2. Fagan's nomogram for FIB-4 >3.85 and mortality.**

mg/L was derived from ROC curve and has sensitivity of 80% and specificity of 75%, PLR of 3.20, NLR of 0.27, and AUC of 0.782 95% CI 0.649–0.914, p = 0.001.

## Multivariable analysis

Multivariate analysis indicated that FIB-4 >3.85 (HR 4.09 95% CI 1.32–12.70, p = 0.015) and CRP >71.57 mg/L (HR 3.36 95% CI 1.08–10.50, p = 0.037) were independently associated with shorter time-to-mortality [Table 2]. Hemoglobin <10 g/dL was not statistically significant in the multivariable model.

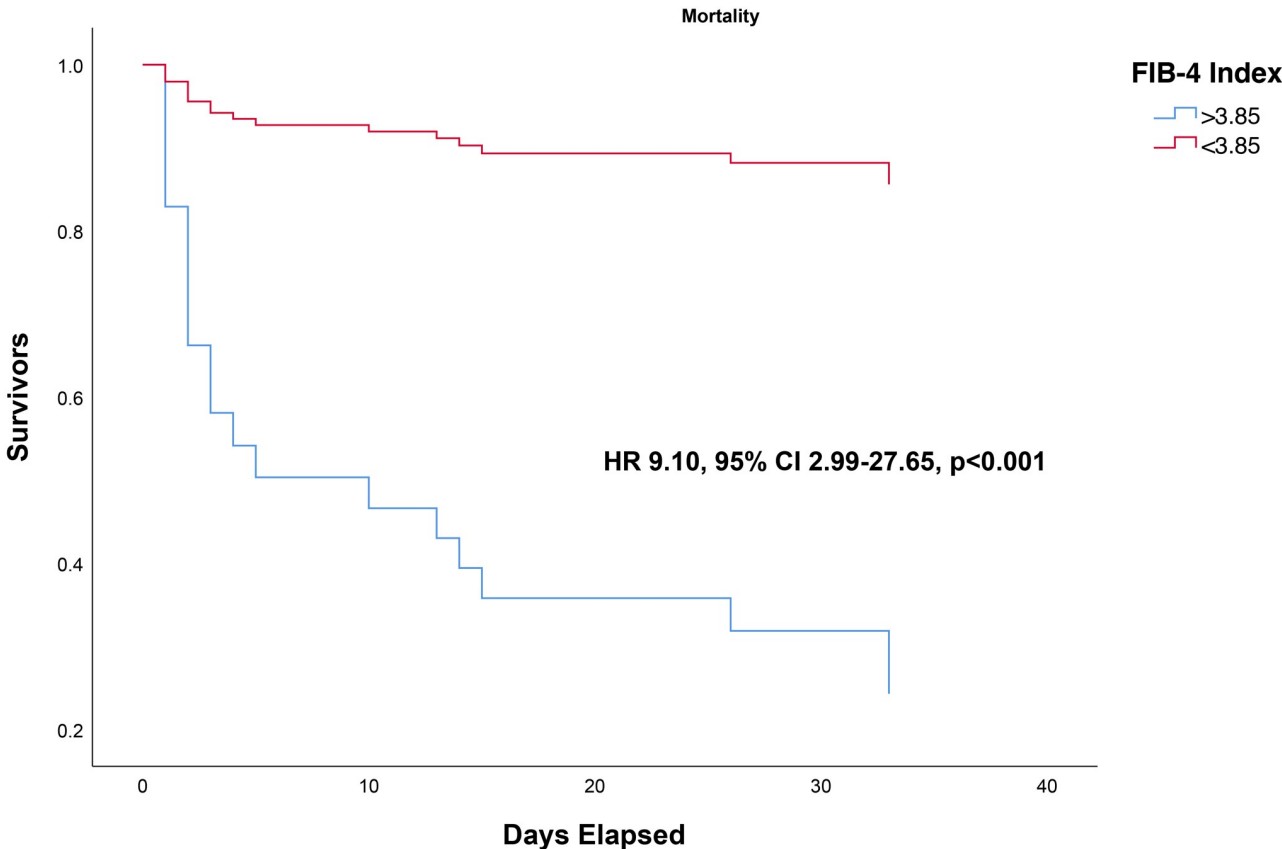

**Fig 3. Time-to-event analysis for FIB-4 >3.85 and mortality (3-months survival).** All of the events occurred at or before the 34[th] day.

## Subgroup analysis based on acute and chronic types

FIB-4 >3.85 was associated with mortality in both acute (OR 8.77 95% CI 1.24–62.25, p = 0.018) and chronic (OR 6.67 95% CI 2.35–18.87, p<0.001) types of malignancies.

This study indicates that a FIB-4 index >3.85 was independent predictor of mortality in patients with hematological malignancies and COVID-19 infection. Subgroup analysis for acute and chronic types of malignancies indicates that the index was applicable in both types of malignancies. Only ORs were obtained for subgroup analysis, because subgroup analysis for HR resulted in spuriously high HR for chronic type of malignancies. FIB-4 >3.85 was significantly associated with mortality in all of these analyses, OR and HR, in both acute and chronic types of malignancies.

Although initially used as a marker of liver fibrosis, FIB-4 index has been repurposed for prognostication in patients with COVID-19, with or without liver disease [7, 9–11]. Age, liver injury, and thrombocytopenia have been shown to increase mortality in COVID-19 [1, 5, 6, 12]. However, in Table 1, only platelets is significantly lower in patients with COVID-19. Although the mean age and median AST ALT were higher in non-survivors, they were not statistically significant. It is possible that due to small sample size, the differences were not noted. This further emphasizes the importance of prediction tool that is more sensitive and specific, combining these factors into FIB-4 index may enhance its performance. FIB-4 index above 3.85 has 79% sensitivity, 84% specificity resulting in 80% posterior probability for mortality in this study. In the multivariate cox-regression model, only FIB-4 >3.85 and CRP >71.57 mg/L

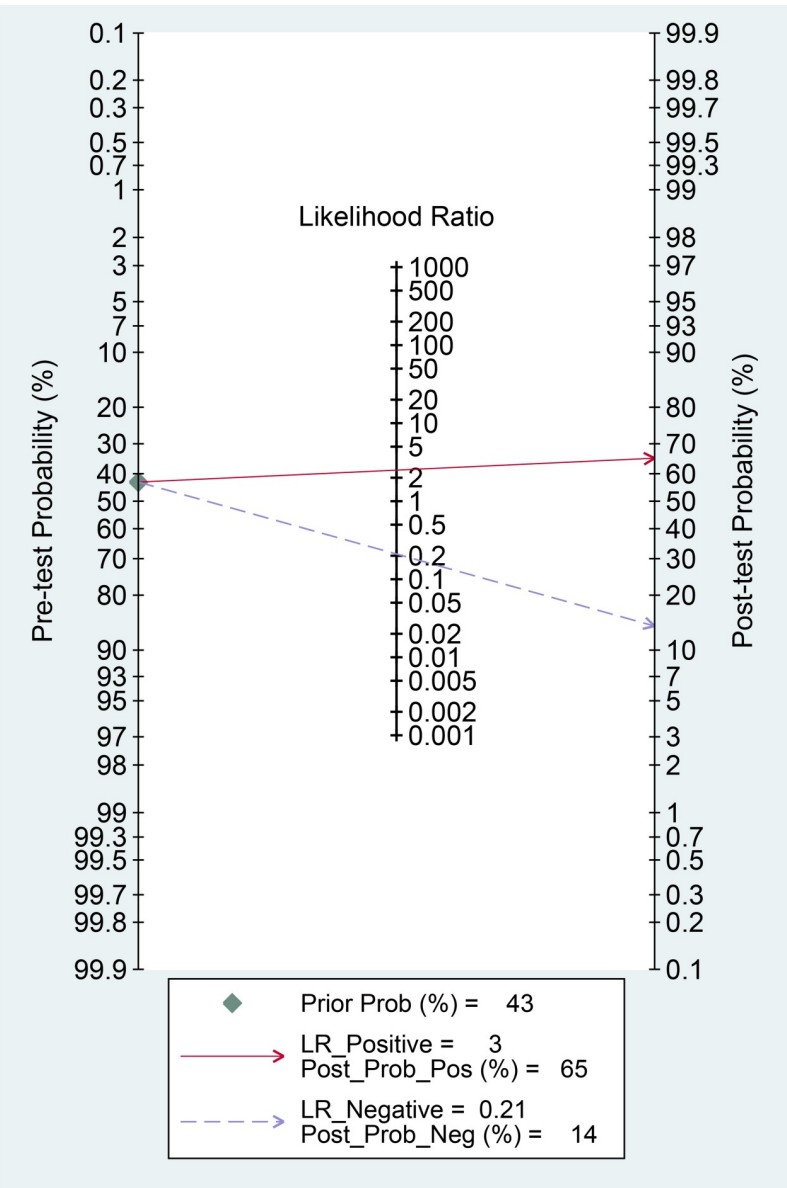

**Fig 4. Fagan's nomogram for FIB-4 >1.45 and mortality.**

independently predicted mortality; with FIB-4 being a stronger predictor. Pro-inflammatory markers such as CRP have been previously shown to be associated with mortality [13]. FIB-4 >1.45 has 84% sensitivity and 59% specificity, thus it may be used to rule-out the poor prognosis. However, other clinical parameters should be considered.

Platelets are an important component of FIB-4 index. COVID-19 induced thrombocytopenia may be caused by altered platelet production and accelerated consumption/destruction [14]. Through the CD13 receptor, SARS-CoV-2 may invade bone marrow cells and platelets which results in growth inhibition and apoptosis [15–17]. Inflammation cytokines may cause inhibition of the hematopoietic stem cells, suppressed thrombopoietin production, and megakaryocyte maturation [18]. Additionally, supplementary hematopoietic progenitor in the pulmonary vessels was reduced by the lung damage associated with COVID-19 [19]. Accelerated

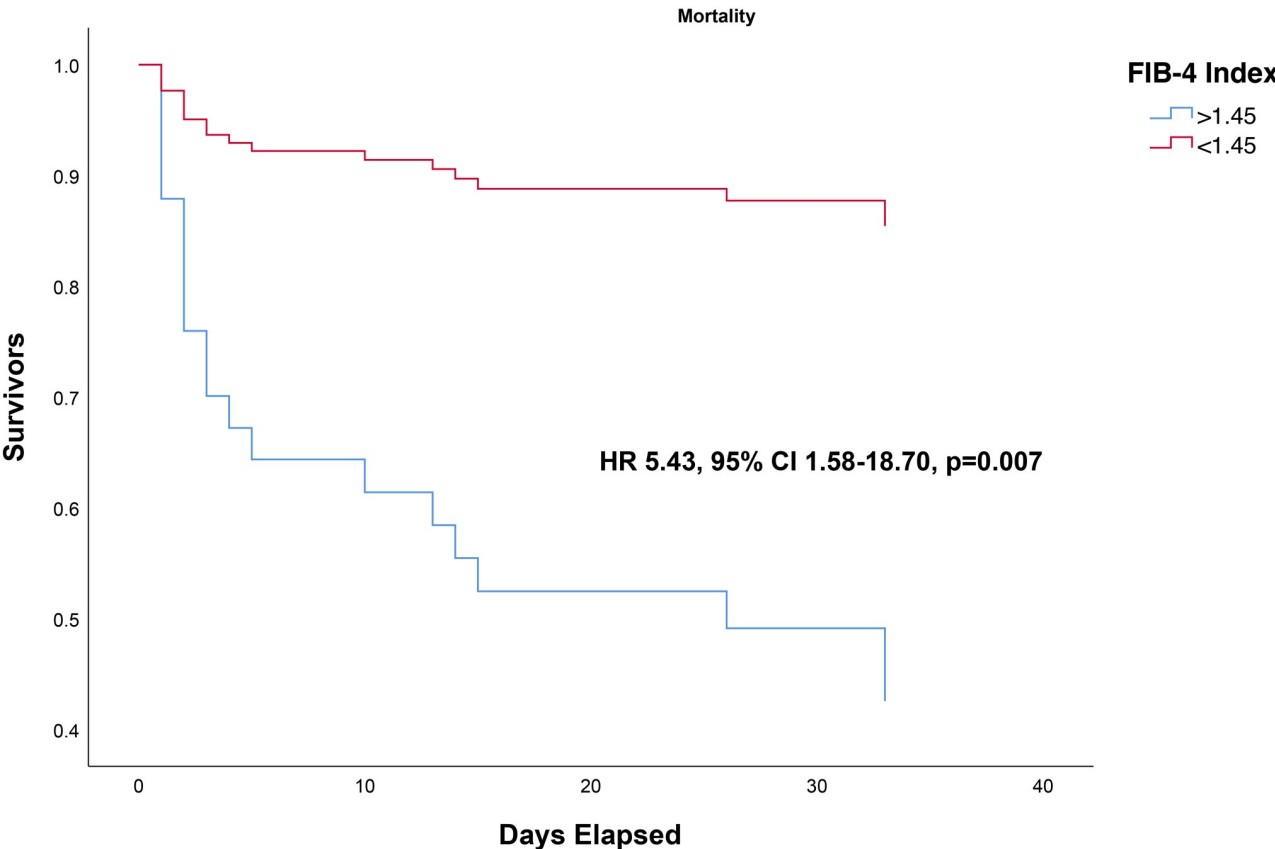

**Fig 5. Time-to-event analysis for FIB-4 >1.45 and mortality (3-months survival).** All of the events occurred at or before the 34[th] day.

platelet consumption or destruction due to inflammation and coagulopathy may further reduce platelets [14]. Another important condition is secondary hemophagocytic lymphohistiocytosis, in which widespread engulfment of blood cells occurred, further reducing platelets [16, 20]. Through molecular mimicry, antibodies generated by the host may specifically bind to platelet antigens causing destruction of platelets [15, 16, 21]. In brief, thrombocytopenia may reflect a higher inflammatory response and greater viral load of SARS-CoV-2, signifying poor prognosis. Thus, COVID-19 synergizes with the inherent hematopoietic abnormalities caused by hematological malignancies, which may further aggravates the disease. In this study, platelets were considerably lower in non-survivors.

Liver enzyme is another important component of FIB-4 Index. Direct viral invasion, medication-induced hepatotoxicity, and inflammation may cause liver injury in patients with COVID-19 [22, 23]. Bile duct epithelial cells and liver endothelial cells express a high amount

**Table 2. The univariable and multivariable cox-regression analysis.**

|  | Unadjusted HR | Adjusted HR |
|---|---|---|
| **FIB-4 >3.85** | 9.10 95% CI 2.99–27.65, p<0.001 | 4.09 95% CI 1.32–12.70, p = 0.015 |
| **Hemoglobin <10 g/dL** | 8.13 95% CI 2.44–27.03, p = 0.001 | 3.65 95% CI 0.82–16.18, p = 0.088 |
| **CRP >71.57 (mg/L)** | 4.98 95% CI 1.86–13.3, p = 0.001 | 3.36 95% CI 1.08–10.50, p = 0.037 |

CRP: c-reactive protein, FIB-4 index: fibrosis-4 index; HR: hazard ratio

of angiotensin-converting enzyme II (ACE2) which is the target for SARS-CoV-2 spike protein [24]. This resulted in increased AST and ALT, followed by increased bilirubin to some extent [25–31]. Most cases are transient and reversible, nevertheless some developed advanced liver injury [24, 32]. In patients with hematological malignancies, increased liver enzyme might be caused by drug toxicity or leukemic cell invasion [33].

## Limitations

This study has several caveats. Despite being the national referral center and the largest hospital for cancer, the sample size is small, nevertheless this study represents the real world data. We are unable to provide stratified analysis based on the types of hematological malignancies or their properties such as resistance/relapse due to small sample size. Additional inflammatory markers such as interleukins were not measured. It would be interesting to see how FIB-4 index fare compared to these variables. Finally, the reasoning for the formula is because one of the authors (Pranata R) published a meta-analysis on the use of FIB-4 Index in patients with COVID-19 [8]. Many of the studies included in the meta-analysis were conducted in patients without liver diseases and FIB-4 Index was shown to be associated with mortality. Thus this is a simple repurposing of a widely used scoring method which include several important variables in patients with hematological malignancies. Familiar score that has been used widely can be applied with relative ease to clinical practice. The model is not based on meticulous formula derived from rigorous evaluation. Since the study is based on limited sample size, this study serves as a basis for further investigation, it will be interesting if other studies in the future are conducted to confirm or debate the findings of the present study. FIB-4 index is yet to be investigated in hematological malignancies without COVID-19, it will be interesting to know how this index fare in future studies.

## Conclusion

FIB-4 index was independently associated with mortality in patients with hematological malignancies and COVID-19. A FIB-4 index cut-off value of 3.85 is optimal for predicting mortality.

## Supporting information

**S1 File.**
(DOCX)

## Acknowledgments

The COVID-19 mitigation team; Department of research and development, Dharmais National Cancer Center, Jakarta, Indonesia.

## Author Contributions

**Conceptualization:** Noorwati Sutandyo, Raymond Pranata.

**Data curation:** Noorwati Sutandyo, Sri Agustini Kurniawati, Achmad Mulawarman Jayusman, Anisa Hana Syafiyah, Arif Riswahyudi Hanafi.

**Formal analysis:** Raymond Pranata.

**Investigation:** Noorwati Sutandyo, Sri Agustini Kurniawati, Achmad Mulawarman Jayusman, Raymond Pranata.

**Methodology:** Raymond Pranata.

**Software:** Raymond Pranata.

**Supervision:** Noorwati Sutandyo.

**Validation:** Raymond Pranata.

**Writing – original draft:** Noorwati Sutandyo, Anisa Hana Syafiyah, Raymond Pranata.

**Writing – review & editing:** Noorwati Sutandyo, Sri Agustini Kurniawati, Achmad Mulawarman Jayusman, Raymond Pranata, Arif Riswahyudi Hanafi.

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
