## [Decision Letter · Decision Letter 0]

21 Jun 2021

PONE-D-21-11387

Repurposing FIB-4 Index as a Predictor of Mortality in Patients with Hematological Malignancies and COVID-19

PLOS ONE

Dear Dr. Sutandyo,

Thank you for submitting your manuscript to PLOS ONE. After careful consideration, we feel that it has merit but does not fully meet PLOS ONE’s publication criteria as it currently stands. Therefore, we invite you to submit a revised version of the manuscript that addresses the points raised during the review process.

We have revived the opinions of expert reviewers as agree with reviewers comments raised a few concerns about this study. We invite you to submit a revised version of the manuscript, please consider and address each of the comments raised by the reviewers.  

We look forward to receiving your revised manuscript.

Kind regards,

Senthilnathan Palaniyandi, Ph.D

Academic Editor

PLOS ONE

Journal Requirements:

2. If you are reporting a retrospective study of medical records or archived samples, please provide additional information about the patient records used in your study in the ethics statement in the manuscript and in the online submission form. Specifically, please ensure that you have discussed whether all data were fully anonymized before you accessed them and/or whether the IRB or ethics committee waived the requirement for informed consent. If patients provided informed written consent to have data from their medical records used in research, please include this information.

4. In your Data Availability statement, you have not specified where the minimal data set underlying the results described in your manuscript can be found. PLOS defines a study's minimal data set as the underlying data used to reach the conclusions drawn

 in the manuscript and any additional data required to replicate the reported study findings in their entirety. All PLOS journals require that the minimal data set be made fully available. For more information about our data policy, please see http://journals.plos.org/plosone/s/data-availability.

"Upon re-submitting your revised manuscript, please upload your study’s minimal underlying data set as either Supporting Information files or to a stable, public repository and include the relevant URLs, DOIs, or accession numbers within your revised

 cover letter. For a list of acceptable repositories, please see http://journals.plos.org/plosone/s/data-availability#loc-recommended-repositories. Any potentially identifying patient information must be fully anonymized.

Important: If there are ethical or legal restrictions to sharing your data publicly, please explain these restrictions in detail. Please see our guidelines for more information on what we consider unacceptable restrictions to publicly sharing data: http://journals.plos.org/plosone/s/data-availability#loc-unacceptable-data-access-restrictions.

 Note that it is not acceptable for the authors to be the sole named individuals responsible for ensuring data access.

7. Please include your tables as part of your main manuscript and remove the individual files. Please note that supplementary tables (should remain/ be uploaded) as separate "supporting information" files

Reviewers' comments:

Reviewer's Responses to Questions

**Comments to the Author**

1. Is the manuscript technically sound, and do the data support the conclusions?

Reviewer #1: Partly

Reviewer #2: Partly

2. Has the statistical analysis been performed appropriately and rigorously? 

Reviewer #1: Yes

Reviewer #2: No

3. Have the authors made all data underlying the findings in their manuscript fully available?

Reviewer #1: Yes

Reviewer #2: Yes

4. Is the manuscript presented in an intelligible fashion and written in standard English?

Reviewer #1: Yes

Reviewer #2: Yes

5. Review Comments to the Author

Reviewer #1: Knowledge of COVID-19 pathogenesis in cancer patients is very limited but accumulating steadily. Any efforts in the direction are useful and extremely important. Data has to be examined carefully prior to any conclusions considering all the possibilities and probabilities within the scope of the study.

Studies correlating FIB-4 index with COVID-19 infections in the context of fibrosis and type II diabetes milletus have already been published. In the current study authors propose a correlation between FIB-4 index and mortality in hematological malignancies.

In addition to the caveats listed by the authors there are more questions that need to be addressed.

1.Is there correlation between COVID -19 negative or low load infection to heavy load of the infection in cancer patients.

2. Are there any insights in to possible treatment regimen so as to reduce the viral load and not aggravate cancer progression.

3. The sample size is limited. It would be interesting to observe data from multiple studies to be able to establish a correlation between FIB-4 index and mortality in hematological malignancies and COVID-19.

Reviewer #2: The authors evaluated the prognostic value of FIB-4 in patients with various hematologic malignancies. FIB-4 is calculated by AST, ALT, platelet and age, and was originally developed for the assessment of chronic liver disease. There have been some previous publication highlighting the value of FIB-4 in patients with acute illness like COVID19. The hypothesis is FIB-4 is prognostic in this population. The analysis seems to be reasonable, though the clinically more important question “what is the impact of underlying hematologic malignancy in patients with COVID19” was not assessed in this study, as there is no COVID19 patient data without hematologic malignancy presented.

The table 1 shows grouping of patients only by “survivor” and “non-survivor”. For this kind of analysis, sufficient follow up duration for all patients is important. The authors state the median follow up duration of 33 days but no range. If “mortality” is the endpoint, it is best to provide a specified time mortality (i.e. 30-day mortality etc) using Kaplan-Meyer estimate. The Table 2 is evaluated for “time to mortality”, which seems appropriate.

Remission and relapse information are provided but majority were “neither”. It is not clear what exactly it means. If it indicates that the patient was undergoing the initial therapy for the primary malignancy, it should be clarified. The implication is significantly different if they were on treatment versus not. The research would be valuable if this study was focused on patients who were undergoing treatment for hematologic malignancy or those who experienced COVID during therapy or within 30 (or whatever number) days from the completion of treatment. Or, the complete opposite would be to focus on those who were in remission and not receiving treatment. In any case, such full analyses may not be necessarily feasible due to limited number of patients.

The timing that FIB-4 was calculated based on the value at the time of admission. It is possible that the FIB-4 reflects the severity of the disease as it worsens, but not necessarily prognostic as one single parameter at diagnosis. It is worth mentioning the time from the onset of symptoms to admission (FIB-4 assessment) or time from diagnosis to admission (FIB-4 assessment). Or FIB-4 at the time of diagnosis, if available, would be also valuable.

At last, but not least, FIB-4 formula was “borrowed” from other studies (originally from chronic liver scarring) but there is no rigorous evaluation of the formula itself in this context. Why ALT needs to be square rooted and placed in the denominator? The impact of age (e.g. 1 vs 10 vs 50) is appropriately reflected in this formula? Such discussion or consideration, or to state the limitation of the current study in this regard would strengthen the paper.

6. PLOS authors have the option to publish the peer review history of their article (what does this mean?). If published, this will include your full peer review and any attached files.

Reviewer #1: No

Reviewer #2: No

---

## [Author Response · Author response to Decision Letter 0]

3 Jul 2021

Reviewer #1: Knowledge of COVID-19 pathogenesis in cancer patients is very limited but accumulating steadily. Any efforts in the direction are useful and extremely important. Data has to be examined carefully prior to any conclusions considering all the possibilities and probabilities within the scope of the study.

Studies correlating FIB-4 index with COVID-19 infections in the context of fibrosis and type II diabetes milletus have already been published. In the current study authors propose a correlation between FIB-4 index and mortality in hematological malignancies.

In addition to the caveats listed by the authors there are more questions that need to be addressed.

Response: Thank you very much for reviewing our paper and providing constructive feedbacks for our article.

1.Is there correlation between COVID -19 negative or low load infection to heavy load of the infection in cancer patients.

Response: Thank you very much for the question, we are sure that it will be an interesting analysis. Unfortunately, the data on patients’ viral load are unavailable. Thus we are unable to perform the analysis.

2. Are there any insights in to possible treatment regimen so as to reduce the viral load and not aggravate cancer progression.

Response: Thank you very much for the question, in the Dharmais National Cancer hospital of Indonesia, we used 

1. Avigan (Favipiravir) 2 x 1600 mg followed by 2 x 600 mg maintenance

2. Azithromycin 1 x 500 mg

3. Oseltamivir 2x75 mg

4. Vitamin D3 1x1000 IU

5. Vitamin C 2x500 mg

6. Zinc 1x20 mg

7. Betadine gargle 3x15 ml

8. Iodine nasal spray 3x1

Theoretically, we observe no potential mechanisms or drug-to-drug interaction that may aggravate cancer progression with the use of these medications. Although control group with other medications are required to sufficiently conclude.

3. The sample size is limited. It would be interesting to observe data from multiple studies to be able to establish a correlation between FIB-4 index and mortality in hematological malignancies and COVID-19.

Response: Thank you very much for the suggestion, to the best of the authors’ knowledge, this is the first study evaluating the use of FIB-4 index as prognostication tool in patients with hematological malignancies with COVID-19. There are others in different type of population. Thus, this study serve as a basis for further investigation, it will be interesting if other studies in the future are conducted to confirm or debate the findings of the present study. We have added this in the limitation section.

Reviewer #2: The authors evaluated the prognostic value of FIB-4 in patients with various hematologic malignancies. FIB-4 is calculated by AST, ALT, platelet and age, and was originally developed for the assessment of chronic liver disease. There have been some previous publication highlighting the value of FIB-4 in patients with acute illness like COVID19. The hypothesis is FIB-4 is prognostic in this population. The analysis seems to be reasonable, though the clinically more important question “what is the impact of underlying hematologic malignancy in patients with COVID19” was not assessed in this study, as there is no COVID19 patient data without hematologic malignancy presented.

Response: Thank you very much for reviewing our paper and providing constructive feedbacks for our article.

The table 1 shows grouping of patients only by “survivor” and “non-survivor”. For this kind of analysis, sufficient follow up duration for all patients is important. The authors state the median follow up duration of 33 days but no range. If “mortality” is the endpoint, it is best to provide a specified time mortality (i.e. 30-day mortality etc) using Kaplan-Meyer estimate. The Table 2 is evaluated for “time to mortality”, which seems appropriate.

Response: Thank you very much for the suggestion, the patients were followed-up for 90 days and the Table 1 is for 3-months survival, we have added details in the methods and table. However, the event (mortality) occurred from one to 34 days. The Kaplan Meier estimate was for 90 days, however, the graph ended at the 34th day since all of the events occurred at or before the 34th day; we have revised the figure caption to add the details.

Remission and relapse information are provided but majority were “neither”. It is not clear what exactly it means. If it indicates that the patient was undergoing the initial therapy for the primary malignancy, it should be clarified. The implication is significantly different if they were on treatment versus not. The research would be valuable if this study was focused on patients who were undergoing treatment for hematologic malignancy or those who experienced COVID during therapy or within 30 (or whatever number) days from the completion of treatment. Or, the complete opposite would be to focus on those who were in remission and not receiving treatment. In any case, such full analyses may not be necessarily feasible due to limited number of patients.

Response: Thank you very much for the question, this study is conducted in patients undergoing chemotherapy for the primary malignancy. Thus all patients are currently receiving chemotherapy. We have added this to the methods section, we would like to apologize for the vague statement.

The timing that FIB-4 was calculated based on the value at the time of admission. It is possible that the FIB-4 reflects the severity of the disease as it worsens, but not necessarily prognostic as one single parameter at diagnosis. It is worth mentioning the time from the onset of symptoms to admission (FIB-4 assessment) or time from diagnosis to admission (FIB-4 assessment). Or FIB-4 at the time of diagnosis, if available, would be also valuable.

Response: Thank you very much for the suggestion, all patients in this study are patients undergoing chemotherapy for hematological malignancies. COVID-19 screening test using RT-PCR of nasopharyngeal samples are performed prior to hospitalization for chemotherapy, thus, the time from diagnosis to admission is within a few hours. We have added this to the methods section. As for the onset of symptoms to admission, we cannot provide adequate/clear documentation because many of symptoms overlap with those of malignancies and many patients display no COVID-19 related symptoms (24 patients).

At last, but not least, FIB-4 formula was “borrowed” from other studies (originally from chronic liver scarring) but there is no rigorous evaluation of the formula itself in this context. Why ALT needs to be square rooted and placed in the denominator? The impact of age (e.g. 1 vs 10 vs 50) is appropriately reflected in this formula? Such discussion or consideration, or to state the limitation of the current study in this regard would strengthen the paper.

Response: Thank you very much for the suggestion. Unfortunately, the reasoning for the formula is because one of the authors (Pranata R) published a paper on the use of FIB-4 Index in patients with COVID-19 (meta-analysis), many of the included studies are conducted in patients without liver diseases. Thus this is a simple repurposing of a widely used scoring method (so it can be directly used without creating an additional model to an already congested pool of prediction models), which include several important variables in patients with hematological malignancies. It turns out to be a good prediction model. Thus the model is not based on meticulous formula derived from rigorous evaluation, we have added this to the limitation section.

---

## [Decision Letter · Decision Letter 1]

2 Sep 2021

PONE-D-21-11387R1

Repurposing FIB-4 Index as a Predictor of Mortality in Patients with Hematological Malignancies and COVID-19

PLOS ONE

Dear Dr. Sutandyo,

Thank you for submitting your manuscript to PLOS ONE. After careful consideration, we feel that it has merit but does not fully meet PLOS ONE’s publication criteria as it currently stands. Therefore, we invite you to submit a revised version of the manuscript that addresses the points raised during the review process.

We have received the opinions of expert reviewer and we invite you to submit a revised version of the manuscript. 

We look forward to receiving your revised manuscript.

Kind regards,

Senthilnathan Palaniyandi, Ph.D

Academic Editor

PLOS ONE

Journal Requirements:

Additional Editor Comments (if provided):

Reviewers' comments:

Reviewer's Responses to Questions

**Comments to the Author**

1. If the authors have adequately addressed your comments raised in a previous round of review and you feel that this manuscript is now acceptable for publication, you may indicate that here to bypass the “Comments to the Author” section, enter your conflict of interest statement in the “Confidential to Editor” section, and submit your "Accept" recommendation.

Reviewer #2: All comments have been addressed

2. Is the manuscript technically sound, and do the data support the conclusions?

Reviewer #2: Yes

3. Has the statistical analysis been performed appropriately and rigorously? 

Reviewer #2: Yes

4. Have the authors made all data underlying the findings in their manuscript fully available?

Reviewer #2: Yes

5. Is the manuscript presented in an intelligible fashion and written in standard English?

Reviewer #2: Yes

6. Review Comments to the Author

Reviewer #2: The authors addressed points that were raised by the reviewers adequately.

The only minor point is, in Page 8, "Consecutive sampling of adults who were tested positive for COVID19 while undergoing chemotherapy for hematological malignancy" would be a better description of the population analyzed.

7. PLOS authors have the option to publish the peer review history of their article (what does this mean?). If published, this will include your full peer review and any attached files.

Reviewer #2: No

---

## [Author Response · Author response to Decision Letter 1]

4 Sep 2021

Reviewer #2: The authors addressed points that were raised by the reviewers adequately.

The only minor point is, in Page 8, "Consecutive sampling of adults who were tested positive for COVID19 while undergoing chemotherapy for hematological malignancy" would be a better description of the population analyzed.

Response: Thank you very much for the suggestion, we have revised accordingly.

---

## [Decision Letter · Decision Letter 2]

10 Sep 2021

Repurposing FIB-4 Index as a Predictor of Mortality in Patients with Hematological Malignancies and COVID-19

PONE-D-21-11387R2

Dear Dr. Sutandyo,

We’re pleased to inform you that your manuscript has been judged scientifically suitable for publication and will be formally accepted for publication once it meets all outstanding technical requirements.

Kind regards,

Senthilnathan Palaniyandi, Ph.D

Academic Editor

PLOS ONE

Additional Editor Comments (optional):

Reviewers' comments:

Reviewer's Responses to Questions

**Comments to the Author**

1. If the authors have adequately addressed your comments raised in a previous round of review and you feel that this manuscript is now acceptable for publication, you may indicate that here to bypass the “Comments to the Author” section, enter your conflict of interest statement in the “Confidential to Editor” section, and submit your "Accept" recommendation.

Reviewer #2: All comments have been addressed

2. Is the manuscript technically sound, and do the data support the conclusions?

Reviewer #2: Yes

3. Has the statistical analysis been performed appropriately and rigorously? 

Reviewer #2: Yes

4. Have the authors made all data underlying the findings in their manuscript fully available?

Reviewer #2: No

5. Is the manuscript presented in an intelligible fashion and written in standard English?

Reviewer #2: Yes

6. Review Comments to the Author

Reviewer #2: All points have been addressed.

7. PLOS authors have the option to publish the peer review history of their article (what does this mean?). If published, this will include your full peer review and any attached files.

Reviewer #2: No

---

## [Editor Report · Acceptance letter]

14 Sep 2021

PONE-D-21-11387R2 

Repurposing FIB-4 Index as a Predictor of Mortality in Patients with Hematological Malignancies and COVID-19 

Dear Dr. Sutandyo:

I'm pleased to inform you that your manuscript has been deemed suitable for publication in PLOS ONE. Congratulations! Your manuscript is now with our production department. 

Kind regards, 

on behalf of

Dr. Senthilnathan Palaniyandi 

Academic Editor

PLOS ONE